# Influence of Hospital Bed Count on the Positioning of Cardiovascular Interventional Radiology (IR) Nurses: Online Questionnaire Survey of Japanese IR-Specialized Radiological Technologists

**DOI:** 10.3390/nursrep15010011

**Published:** 2025-01-04

**Authors:** Tomoko Kuriyama, Takashi Moritake, Go Hitomi, Koichi Nakagami, Koichi Morota, Satoru Matsuzaki, Hajime Sakamoto, Kazuma Matsumoto, Mamoru Kato, Hiroko Kitamura

**Affiliations:** 1Department of Occupational and Community Health Nursing, School of Health Sciences, University of Occupational and Environmental Health, Japan, Kitakyushu 807-8555, Japan; kuritomo@med.uoeh-u.ac.jp; 2Department of Radiation Regulatory Science Research, Institute for Radiological Science, National Institutes for Quantum Science and Technology, Chiba 263-8555, Japan; matsuzaki.satoru@qst.go.jp; 3Department of Radiological Technology, Kawasaki Medical School Hospital, Kurashiki 701-0192, Japan; hitomi@med.kawasaki-m.ac.jp; 4Department of Radiology, Hospital of the University of the Occupational and Environmental Health, Japan, Kitakyushu 807-8556, Japan; nakagami@clnc.uoeh-u.ac.jp; 5Department of Radiology, Shinkomonji Hospital, Kitakyushu 800-0057, Japan; morota@shinkomonji-hp.jp; 6Department of Radiological Technology, Faculty of Health Science, Juntendo University, Tokyo 113-8421, Japan; h.sakamoto.qv@juntendo.ac.jp; 7Department of Radiological Technology, Hyogo Medical University Hospital, Nishinomiya 663-8501, Japan; matsumo@hyo-med.ac.jp; 8Department of Radiology and Nuclear Medicine, Akita Cerebrospinal and Cardiovascular Center, Akita 010-0874, Japan; kato-mamoru@akita-noken.jp; 9Occupational Health Training Center, University of Occupational and Environmental Health, Japan, Kitakyushu 807-8555, Japan; h-kita@med.uoeh-u.ac.jp

**Keywords:** cardiovascular interventional radiology, radiation protection for IR nurses, position, staff composition, number of hospital beds, risk assessment

## Abstract

Background/Objectives: Interventional radiology (IR) utilizing X-rays can lead to occupational radiation exposure, posing health risks for medical personnel in the field. We previously conducted a survey on the occupational radiation exposure of IR nurses in three designated emergency hospitals in Japan. Our findings indicated that a hospital with 214 beds showed a higher lens-equivalent dose than hospitals with 678 and 1182 beds because the distance between the X-ray irradiation field and the IR nurse’s position of the hospital with 214 beds was shorter than those of 678 and 1182 beds. Based on these observations, we hypothesized that the number of hospital beds affects the distance between the X-ray irradiation field and the IR nurse’s position. Methods: To verify this hypothesis, we conducted a more extensive online questionnaire survey, focusing exclusively on hospitals that perform cardiovascular IR. Results: We analyzed data from 78 facilities. The results of this study confirmed our earlier findings, showing that both the number of physicians performing IR procedures and the distance from the X-ray irradiation field to the IR nurse’s position are influenced by the number of hospital beds. Additionally, factors such as the type of hospital, emergency medical system, annual number of IR sessions, location of medical equipment, and the positioning of IR nurses appear to be associated with the number of hospital beds. Conclusions: Understanding these relationships could enable the development of individualized and prioritized radiation exposure reduction measures for IR nurses in high-risk settings, provided that comprehensive occupational radiation risk assessments for cardiovascular IR consider the number of hospital beds and related factors. This study was not registered.

## 1. Introduction

Interventional radiology (IR) has witnessed significant advancements in imaging technology and device enhancements, enabling complex procedures to be performed with reduced invasiveness. This progress has contributed to a 5.4-fold increase in angiography procedures in Japan, rising from 183,000 in 1993 to 991,000 in 2020 [1]. While medical radiation offers substantial benefits to patients, it also poses health risks in the form of occupational exposure to medical staff involved in these procedures [2]. Consequently, IR, which frequently involves X-ray fluoroscopic procedures, is classified as a hazardous occupation.

During typical fluoroscopic working conditions, the cumulative exposure of interventionalists, other physicians, and/or medical staff working close to the patient, even if the exposure dose per procedure is low, can be high due to repeated procedures, and there is a concern that the risk of radiation injuries will increase. There have been many reports of the high risk of developing lens opacity before cataracts among medical staff working in IR [3,4,5,6,7,8,9,10,11,12]. And now, the threshold dose for the eye lens is thought to be 0.5 Gy [13], and the lens is considered one of the most radiation-sensitive tissues in the human body [14,15].

Our previous survey of occupational exposure among IR nurses at three designated emergency hospitals of varying types and bed counts revealed that none of the IR nurses exceeded the lens equivalent dose limit of 20 mSv/year. However, hospitals with fewer beds tended to have a shorter distance between the X-ray irradiation field and the IR nurse’s position, leading to a higher lens-equivalent dose [16]. Previous studies have also identified factors affecting the radiation dose to IR nurses, including their position relative to the patient and X-ray tube [17,18,19]; their role [19,20,21,22]; the use of shielding devices like ceiling-suspended lead shields (CSS) (Figure 1a) and rolling lead shields (RS) (Figure 1b), to protect physicians and other staff, respectively [19,20,23,24,25]; and their calling out to the operator before approaching the patient [26]. Additionally, since IR nurses do not control the radiation source, it is challenging for them to anticipate their exposure timing [16,27]. To date, our research has not found other studies that specifically examine the impact of hospital bed count on occupational exposure levels of IR nurses.

Thus, this study expanded our prior research and tested the hypothesis from our previous paper [16] that “the number of hospital beds influences the distance between the X-ray irradiation field and the IR nurse’s position”. We conducted an online questionnaire survey to assess whether risk evaluations of radiation exposure to IR nurses could be effectively performed considering the variable of hospital bed numbers.

## 2. Materials and Methods

### 2.1. Study Design

This observational cross-sectional study was conducted using a questionnaire developed by the researcher.

### 2.2. Setting and Context

We carried out a self-administered online survey among IR-specialized radiological technologists working in Japanese hospitals. The survey inquired about the working conditions of IR nurses in angiography rooms at their respective hospitals. Research materials, including a guide for participation, an explanatory document, and the questionnaire, were made available on the researcher’s university website. The participants accessed the relevant website of their own volition. A link to the Japan Professional Accreditation Board of Radiological Technologists for Angiography and Intervention [28] was also provided. Participants were additionally recruited through an email distributed in the newsletter of a related research group certified by the board. The explanatory document on the website was designed to ensure participants’ freedom to participate or refuse, outline the potential benefits and risks of the study, and clarify that the questionnaire would only be distributed to those who consented to participate. The survey was conducted using Google Forms to ensure anonymity and protect the personal information of the participants. The survey was conducted from 1 February 2023 to 30 November 2023.

### 2.3. Participants

Eligible participants were IR-specialized radiological technologists who were currently working or had previously worked in cardiovascular IR.

### 2.4. Data Sources

The questionnaire collected basic information on the affiliated institutions (5 items: multiple choice) and detailed data on the working conditions of cardiovascular IR, focusing on distance and shielding factors that influence radiation exposure (11 items: multiple choice, some written) (Table 1). Residents were defined as physicians licensed for less than five years. The IR nurses included in this survey were circulating nurses, who are responsible for tests, preparing the treatment environment, assisting the patient, and recording information.

The questionnaire was developed by a team of five, including four IR-specialized radiological technologists and one nurse dedicated to researching radiation protection for medical workers. We assessed the stability of the questionnaire’s scale by conducting two pre-tests with a one-month interval to verify test–retest reliability. The kappa coefficient (κ) was used to measure agreement, adjusted for nominal and ordinal scales. The agreement criteria were set as follows: slight agreement (κ = 0.0–0.20), fair agreement (κ = 0.21–0.40), moderate agreement (κ = 0.41–0.60), substantial agreement (κ = 0.61–0.80), and almost perfect or perfect agreement (κ = 0.81–1.00) [29]. Questions scoring below 0.40, such as those related to the proportion of cases involving three or more nurses for diagnostic or therapeutic purposes, were excluded due to low stability.

### 2.5. Statistical Method

Responses were categorized into three groups based on the number of hospital beds and analyzed using the Kruskal–Wallis test. Multiple comparisons were performed using the Mann–Whitney *U* test, with adjustments for multiple testing (Bonferroni adjustment). Hospital characteristics and the conditions within the cardiovascular IR angiography rooms were compared using cross-tabulation and the chi-square (χ^2^) test. The analysis was conducted using SPSS Ver. 25 for Windows (SPSS Inc., Chicago, IL, USA). All *p*-values reported were two-tailed, with statistical significance set at *p* < 0.05.

## 3. Results

We received responses from 82 facilities. After excluding four facilities due to overlap in hospital type, district location, emergency medical system, and number of hospital beds (suggesting they were responses from the same facility), we analyzed data from 78 facilities. This represents approximately 18% of the 434 facilities in Japan where IR-specialized radiological technologists are employed; thus, the data’s sampling error could be estimated to be approximately 10% (confidence level 95%). The demographics of the respondents were predominantly male (89.7%), with 76.9% having at least ten years of experience working in angiography rooms. We received responses from 31 out of 47 prefectures, covering 66% of the total, with a valid response rate of 100%. The median number of hospital beds was 400, ranging from 70 to 1190.

### 3.1. Comparison of Distance Between Position of IR Nurse and X-Ray Irradiation Field by Number of Hospital Beds

The average distance from the center of rotation of the C-arm (X-ray irradiation field) to the location where the IR nurse recorded information (nurse’s position) was 280 cm, with a range of 100–500 cm, and a median [IQR] of 290 220–300 cm. Correlation analysis showed a weak positive correlation (*r* = 0.364) between the number of hospital beds and the distance between the X-ray field and the location of the IR nurse (Figure 2). In hospitals with 521 or more beds, the position of the IR nurse was significantly farther from the X-ray irradiation field compared to hospitals with 520 beds or less (Mann–Whitney *U* test results: ≤343 beds vs. ≥521 beds: *p* = 0.001; 344–520 beds vs. ≥521 beds: *p* = 0.022) (Figure 3 and Appendix A).

### 3.2. Comparison of IR Staff Composition by Number of Hospital Beds

We analyzed differences in IR staff composition relative to the number of hospital beds (Table 2). Generally, most procedures involve two physicians. In detail, hospitals with ≤343 beds often conducted procedures with just one physician, while hospitals with ≥521 beds had significantly fewer procedures performed by a single physician (Mann–Whitney *U*, *p* < 0.05) and a significantly higher proportion of procedures conducted by three or more physicians (Mann–Whitney *U*, *p* < 0.01) compared to hospitals with ≤520 beds. Similar trends were observed in IR for treatment purposes, with almost no procedures conducted by a single physician at any hospital and hospitals with ≥521 beds showing a significantly higher proportion of procedures conducted by three or more physicians (Mann–Whitney *U*, *p* < 0.01). Additionally, the proportion of residents involved in procedures increased with the number of beds, with significantly more residents participating in hospitals with ≥521 beds (Mann–Whitney *U*, IR for diagnosis: *p* < 0.01; IR for treatment: *p* < 0.05).

Regardless of the purpose—diagnostic or treatment—many procedures were conducted by a single IR nurse. The proportion of procedures conducted by only one IR nurse tended to increase with the number of hospital beds, though this difference was not statistically significant.

### 3.3. Comparison of Other Characteristics by Number of Hospital Beds

Characteristics varied according to hospital bed count (χ^2^ test) (Table 3). University hospitals were predominantly found among those with ≥521 beds (*p* = 0.017). In terms of emergency medical system types, primary (initial) emergency care was more common in hospitals with ≤343 beds, secondary emergency care was more frequent in hospitals with 344–520 beds, and tertiary emergency care (emergency medical centers) was more prevalent in hospitals with ≥521 beds (*p* < 0.001). Regarding the annual number of cardiovascular IR procedures, hospitals with ≤343 beds mostly conducted fewer than 300 cases, whereas those with ≥521 beds often conducted 700 or more cases (*p* < 0.001). We received responses from 55 facilities about the size of the angiography room, but no significant differences were found based on the hospital bed count. Hospitals with ≥521 beds were likelier to have some medical equipment outside the angiography room (*p* = 0.020). The absence of CSS (Figure 1a) and RS (Figure 1b) was more common in hospitals with ≤343 beds (adjusted *R* > 1.96), although this difference was not statistically significant.

### 3.4. Comparison of Position of IR Nurse by Number of Hospital Beds

Figure 4 illustrates the standard layout of an angiography room used for cardiovascular IR. The position of the IR nurse (A-T) is divided into six directions, as shown in Figure 5. The working directions of the IR nurse are detailed in Figure 6 and Appendix A. In nearly all hospitals (94.9%), the IR nurse was positioned on the right side of the patient. In hospitals with ≤343 beds, the highest proportion of nurses (61.5%) were positioned at the right side of the patient’s head (R-Head), with the proportion decreasing as hospital bed count increased. In hospitals with ≥521 beds, the most common position for nurses was on the right side of the patient’s body (R-Body) at 40.0%, with responses from two facilities (8.0%) indicating that the IR nurse was positioned outside the angiography room.

### 3.5. Factors That Determine the Position of the IR Nurse

Table 4 outlines the factors that determine the position of the IR nurse. Figure 7 provides a detailed view of these factors, showing their orientation from the patient’s perspective.

## 4. Discussion

We previously surveyed three designated emergency hospitals (two university hospitals and one private hospital) of varying functionality and size, where we found that the lens-equivalent dose for IR nurses did not exceed the dose limit of 20 mSv/year at any facility. However, there was a noticeable difference in the lens-equivalent dose between the hospitals [16]. We hypothesized that this variance was due to smaller hospitals conducting IR procedures with a limited staff—often just one physician and one IR nurse—resulting in a substantially shorter distance between the X-ray irradiation field and the nurse’s position, thereby increasing the nurse’s lens-equivalent dose. In this study, we expanded our survey scope to include a larger range of hospitals to further explore how the number of hospital beds influences the distance between the X-ray irradiation field and the IR nurse’s position. Recognizing that many hospitals have multiple angiography rooms with varying interiors such as equipment, layout, and radiation protection depending on the department and procedures used [16], we surveyed solely on cardiovascular IR. This procedure is the most common in Japan, accounting for 61% of all IR procedures [1].

Our findings reveal that in cardiovascular IR, the distance from the X-ray irradiation field to the position of the IR nurse varies with the size of the hospital. Hospitals with ≥521 beds had a statistically significantly greater distance to the IR nurse’s position compared to hospitals with 344–520 beds or ≤343 beds, echoing trends noted in our previous report [16] (Figure 3 and Appendix A). Distance plays a crucial role in radiation protection, as the radiation exposure dose is inversely proportional to the square of the distance from the radiation source. Although we did not measure the lens-equivalent dose for IR nurses in the current study, we speculated that it was likely lower in hospitals with ≥521 beds than in those with fewer beds.

The most common staffing configuration for cardiovascular IR procedures included two physicians and one IR nurse, with the number of hospital beds influencing this composition (Table 2). In hospitals with ≤343 beds, IR procedures for diagnostic purposes often involved only one physician (average scores of 3.7 for ≤343 beds vs. 1.9 for ≥521 beds), resulting in a tendency to have two or more IR nurses due to the fewer physicians. Conversely, in treatment settings, very few hospitals conducted procedures with just one physician. In hospitals with ≥521 beds, it was more common for at least three physicians to participate (average scores of 2.8 for ≤343 beds vs. 4.3 for ≥521 beds). Additionally, the likelihood of residents participating in IR procedures increased with the number of hospital beds, a trend that aligns with expectations.

As the number of hospital beds increased, hospitals were more likely to be university hospitals involved in medical education for students and residents. They were also more likely to function as advanced tertiary emergency centers and to conduct a higher annual number of cardiovascular IR procedures, all of which are logical outcomes (Table 3).

While no differences were observed in the size of the angiography rooms based on the number of hospital beds, hospitals with ≥521 beds were more likely to lack sufficient storage space for all necessary medical equipment for cardiovascular IR procedures within the angiography room, necessitating some equipment to be stored outside (Table 3). This situation is likely because these larger hospitals are tasked with providing more advanced medical care, including cardiovascular IR. Such treatments require a greater variety and quantity of equipment and more personnel, including residents, which may exceed the capacity of existing angiography rooms. Storing medical equipment outside the angiography room can be advantageous for radiation protection, particularly for rolling nurses who manage supplies and handle patient samples, as it reduces their time in the radiation exposure area. However, this setup tends to have more disadvantages than advantages. One significant concern is the potential for increased scattered radiation leakage outside the angiography room. The frequent opening and closing of the door to move items in and out can lead to increased leakage, particularly if the door’s automatic function fails due to wear, potentially raising the radiation levels in control areas above the limit set at ≤1.3 mSv per three months in Japan [30]. Another issue is maintaining a sterile environment within the angiography room. While many hospitals conduct IR procedures in non-sterile areas, it is advisable for cardiovascular IR procedures to occur in clean areas similar to operating rooms, which necessitates minimizing unnecessary traffic in and out of the angiography room [31]. The recent introduction of hybrid operating rooms, which facilitate IR procedures in a clean environment akin to operating rooms, represents an ideal setup. Improving the conditions for IR nurses in terms of both cleanliness and radiation protection would ideally involve ensuring that the angiography room is large enough to house all necessary equipment and allow nurses to remain inside at all times for efficient management and supply handling [32]. Additionally, the appropriate installation of RS is crucial to provide comprehensive radiation protection for the staff.

In this study, we found that external factors such as the arrangement of medical equipment (46.2%) and room size (41.0%) were the most influential in determining the positions of IR nurses (Table 4). When analyzing the positions of IR nurses, whether at the R-Head (Figure 7a), R-Body (Figure 7b), or R-Foot (Figure 7c) directions, these external factors were consistently the primary determinants. The number of hospital beds also influences the positions of IR nurses, with hospitals with fewer beds typically positioning nurses at the R-Head and with hospitals with a larger number of beds more commonly positioning them at the R-Body (Figure 6). However, room size, another external factor, did not vary with the number of hospital beds (Table 3), indicating that the shift in IR nurses’ positions is likely influenced more by the arrangement of medical equipment and other internal factors. We will further discuss what these internal factors might be, depending on the IR nurses’ positions.

Cardiovascular IR is often performed under local anesthesia with the patient conscious, requiring the IR nurse to monitor the patient’s complaints and facial expressions, respond to any abnormalities, and provide a reassuring environment where patients can comfortably express concerns [33]. The C-arm of the angiography device is positioned in the R-Head direction (Figure 4), with the CSS installed between the X-ray field and the physician, typically leaving the physician out of the R-Head area. This setup, providing more space, allows IR nurses in the R-Head direction to easily observe and communicate with the patient. Additionally, the fifth most cited factor for choosing this position was “ease of assisting the physician” (18.8%) (Figure 7a). The R-Head location, directly to the left of the physician, facilitates the physician’s assistance. Hospitals with ≤343 beds typically have fewer physicians than hospitals with ≥521 beds, making the R-Head direction particularly convenient for IR nurses to access both the patient and the physician simultaneously. However, the R-Head position lacks a CSS between the X-ray field and the IR nurse (Figure 4), potentially resulting in higher radiation exposure. Research indicates that the radiation dose to the IR staff can be 15 times higher when standing next to the patient rather than behind the CSS or physician, even if the distance from the X-ray source remains the same [34]. Therefore, the R-Head direction is a common position for IR nurses in hospitals (*n* = 32). If an IR nurse operates from this direction, additional radiation shielding, such as an RS or personal protective equipment akin to that worn by physicians, may be necessary to ensure their safety.

Hospitals with ≥521 beds often have IR nurses positioned in the R-Body direction (Figure 6), with the fourth most common reason for this choice being the ease of observing the patient’s face (33.3%) (Figure 7b). The R-Body direction places the IR nurse behind the physician, allowing for easier access and a natural shielding effect from the physician’s body positioned between the nurse and the X-ray irradiation field [22]. Although direct communication with the patient is not possible from this position, it allows for clear observation of the patient’s facial expressions. As the number of hospital beds increases, bringing more physicians and limiting space in the R-Head direction, IR nurses may find the R-Body direction to be the next most convenient option. Indeed, the second most common reason for choosing R-Body was the nurse’s opinion (50.0%), and the fifth most common was interdisciplinary discussions between other professions (20.8%) (Figure 7b), suggesting that medical staff view R-Body as a preferred alternative to R-Head for IR nurses.

The R-Foot direction, which does not interfere with angiography equipment, allows IR nurses to easily secure space (Figure 4). However, hospitals with ≤343 beds generally have fewer physicians, resulting in only three hospitals (11.5%) choosing the R-Foot direction over R-Head (Figure 6 and Appendix A). The major drawback of the R-Foot position is the inability to see the patient’s face or communicate directly, and no hospital reported ease of observing the patient’s face as a reason for selecting this direction (Figure 7c). On the other hand, the primary advantage of the R-Foot direction is its distance from the X-ray irradiation field, enhanced by the presence of either a CSS or the physician between the IR nurse and the X-ray source, providing optimal radiation protection. Previous studies have shown that the radiation exposure for staff standing immediately to the right of the first operator is reduced by a quarter, thanks to the protective apron worn by the first operator and their body acting as an effective shield [35]. Correspondingly, reducing radiation exposure was the fourth most cited reason for selecting the R-Foot direction in this survey (33.3%) (Figure 7c).

In a previous survey of three hospitals, we reported that having fewer medical staff involved in IR procedures placed the IR nurse (rolling nurse) closer to the X-ray irradiation field, resulting in higher lens radiation doses for these nurses [16]. We expanded this survey to include 78 hospitals performing cardiovascular IR and found similar results. Although most hospitals have only one IR nurse, the roles assigned to an IR nurse vary. Hospitals with fewer beds tend to employ multiple IR nurses, with physicians likely taking on more direct care responsibilities. In the current study, we examined not only the distance between the X-ray irradiation field and the IR nurse’s position but also the direction of the nurse’s position from the patient’s perspective and the reasons for its selection. Sanchez et al. have shown that the nature of the IR procedure and the required role of the IR nurse significantly influence the nurse’s position and the choice of protective equipment such as RS [19]. Our findings also confirmed that the position of the IR nurse is dictated by the nurse’s role, which varies with the number of hospital beds. Consequently, the relationship between the number of hospital beds and the distance from the X-ray irradiation field to the nurse’s position could be established, making the number of beds a useful metric in assessing IR nurses’ radiation exposure risks. Personal dose monitoring is essential for assessing the exposure risk of IR nurses. However, in the world, there are still many countries and regions where personal dosimeters are unavailable or not worn daily [17,36]. There are also reports stating that the use rate of personal dosimeters is higher among nurses than among physicians [12,37], but in the case of passive dosimeters, the cumulative dose for a certain period is notified to the user, so it is not easy to use the dose results to improve the situation. Even in this situation, the number of hospital beds is a dose indicator that can be obtained easily without spending money, so it is expected to be used for the risk assessment of IR nurses. Looking ahead, further research on radiation exposure risk assessments for IR nurses is necessary. Such studies should aim to enhance the implementation of regular risk assessments, provide targeted radiation protection education, and develop specific strategies for reducing exposure risks among high-risk groups.

There are limitations in the present study. Although we surveyed a significant number of hospitals (78 hospitals), this cannot be considered a trend for all medical facilities in Japan with angiography equipment, of which there are more than 1500. There is a possibility that the responses primarily came from IR-specialized radiological technologists who have a strong interest in radiation protection, which could introduce an online bias. Furthermore, when asking why IR nurses chose their specific working directions, IR-specialized radiological technologists might have selected options that appear more rational in terms of radiation protection compared to their non-specialized counterparts. Therefore, caution should be exercised when interpreting these results.

## 5. Conclusions

In this study, we demonstrated that the most common staffing configuration for cardiovascular IR typically includes two physicians and one IR nurse. We observed that hospitals with a larger number of beds tend to employ more physicians, whereas smaller hospitals may operate with only one physician conducting IR. Additionally, the distance from the X-ray irradiation field to the position of the IR nurse was influenced by the number of hospital beds, with statistically significant longer distances observed in hospitals with ≥521 beds compared to hospitals with ≤520 beds. Furthermore, variations in the number of hospital beds correlate closely with the type of hospital, the emergency medical system, the annual number of IR sessions conducted, the arrangement of medical equipment, and the positioning of IR nurses. Taking these factors into account can enable a more accurate assessment of radiation exposure risks for cardiovascular IR nurses and facilitate effective strategies to reduce radiation exposure, particularly through individualized and prioritized approaches for those at the highest risk.

## Figures and Tables

**Figure 1 nursrep-15-00011-f001:**
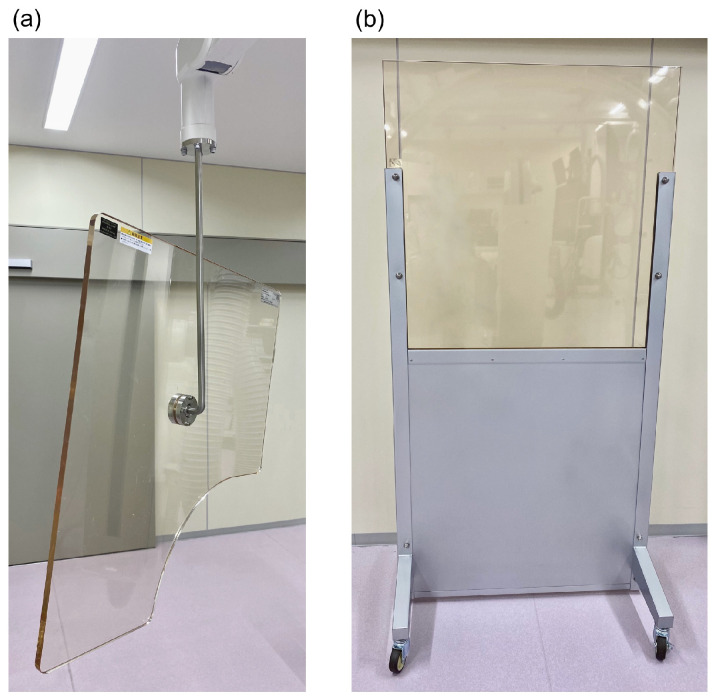
Lead-line radiation-shielding equipment: (**a**) ceiling-suspended lead shield (CSS) and (**b**) rolling lead shield (RS).

**Figure 2 nursrep-15-00011-f002:**
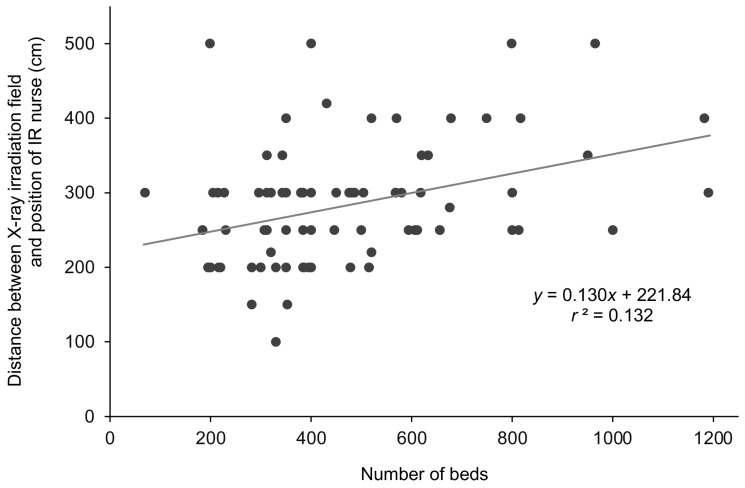
Association between the number of hospital beds and the distance between the X-ray irradiation field and the position of the IR nurse. ● indicates each hospital. The solid line shows the regression line. *r*^2^, coefficient of determination.

**Figure 3 nursrep-15-00011-f003:**
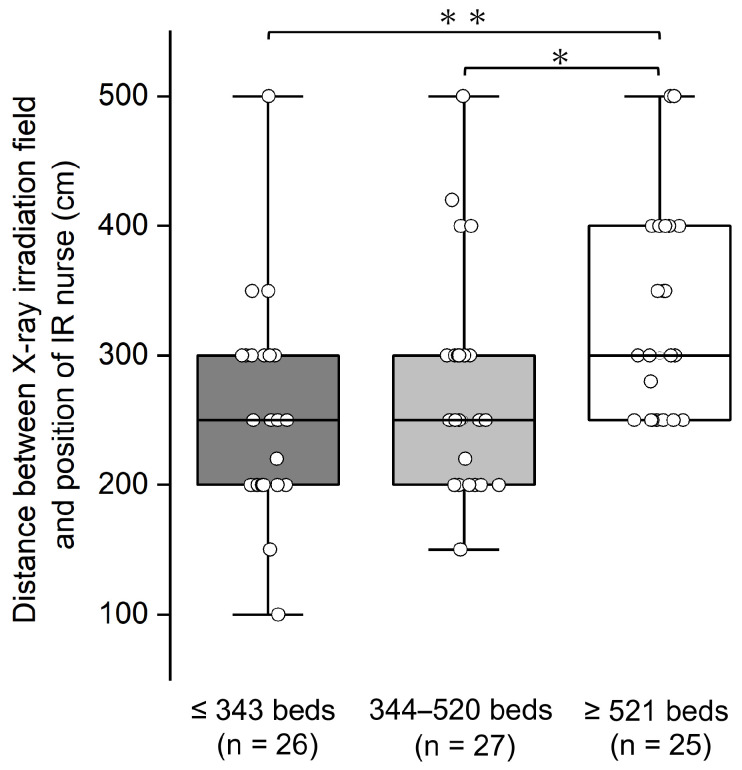
Comparison of the distance between the X-ray irradiation field and the position of the IR nurse based on the number of hospital beds. The bottom and top of the box are the first and third quartiles, and the line inside the box is the second quartile, the median. The bottom and top of the whiskers are the minimum and maximum values. ○ represents each hospital. Mann–Whitney *U* test (Bonferroni-adjusted for multiple comparisons), * *p* < 0.05, ** *p* < 0.01.

**Figure 4 nursrep-15-00011-f004:**
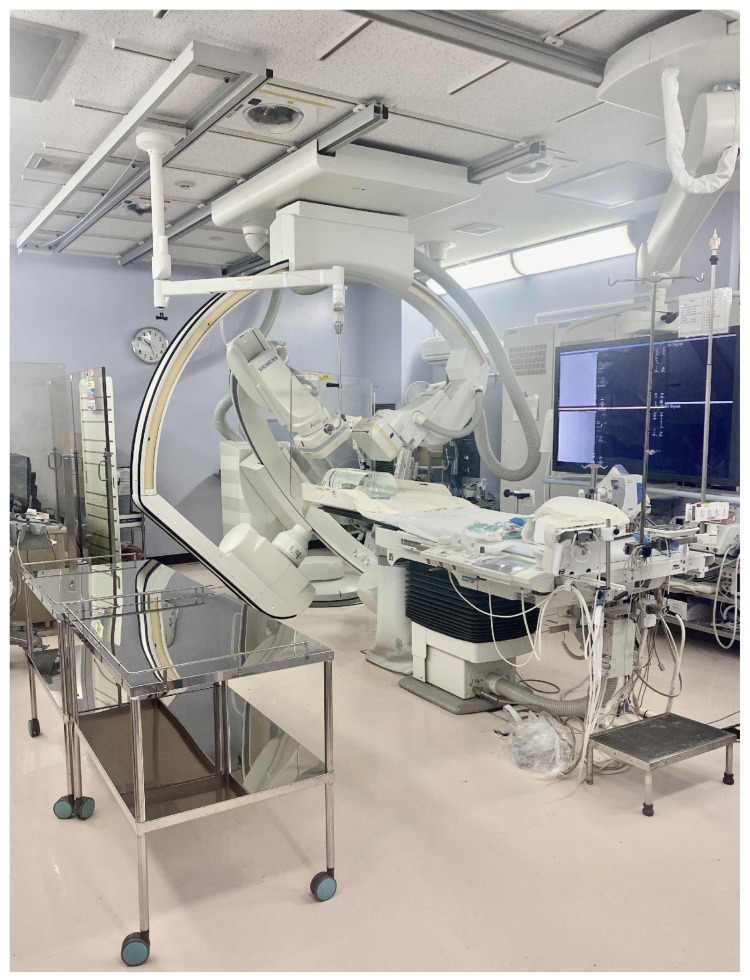
Example layout of an angiography room used in cardiovascular IR.

**Figure 5 nursrep-15-00011-f005:**
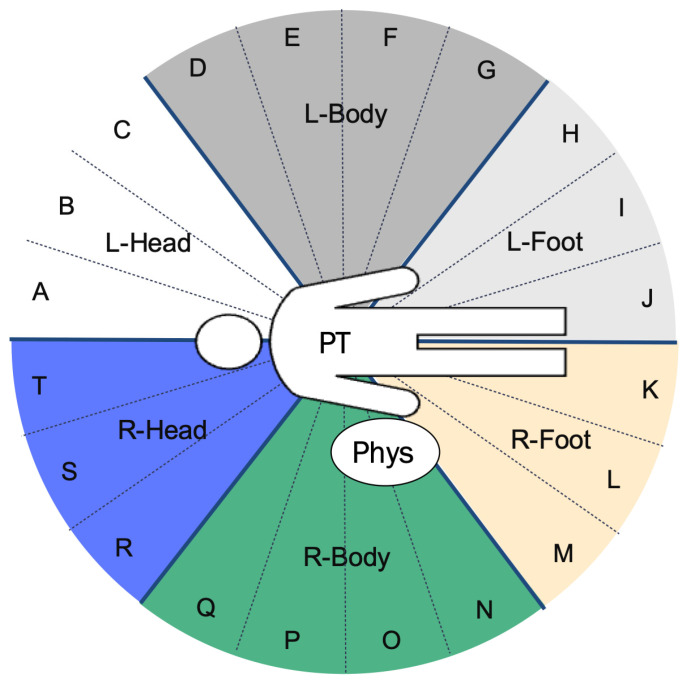
Directions in the angiography room divided into 20 sections as viewed from the patient’s perspective (A-T). These sections are further classified into six directions: on the right of the patient’s head (R-Head, shown in blue), right of the patient’s body (R-Body, shown in green), right of the patient’s foot (R-Foot, shown in ivory), left of the patient’s head (L-Head, shown in white), left of the patient’s body (L-Body, shown in gray), and left of the patient’s foot (L-Foot, shown in light gray). “Phys” indicates the position of the main physician. “PT” denotes the patient.

**Figure 6 nursrep-15-00011-f006:**
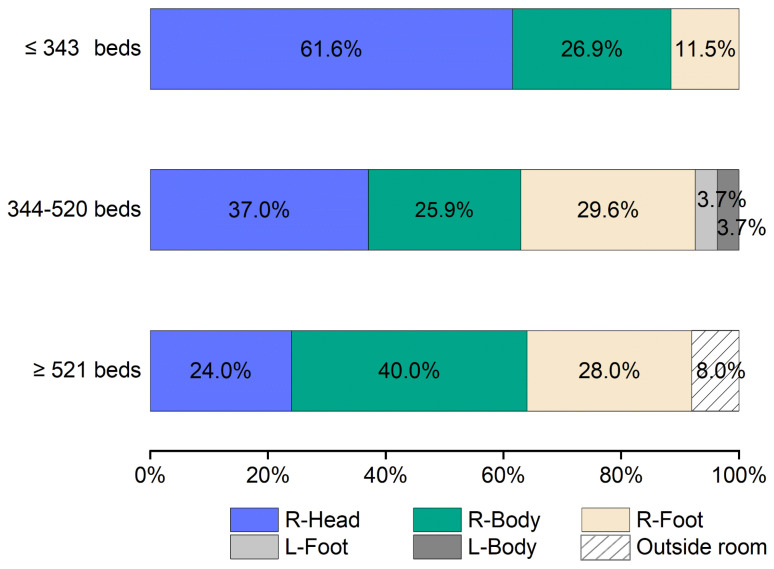
Positions (directions) of cardiovascular IR nurses by the number of hospital beds. Note: No facilities selected the L-Head direction.

**Figure 7 nursrep-15-00011-f007:**
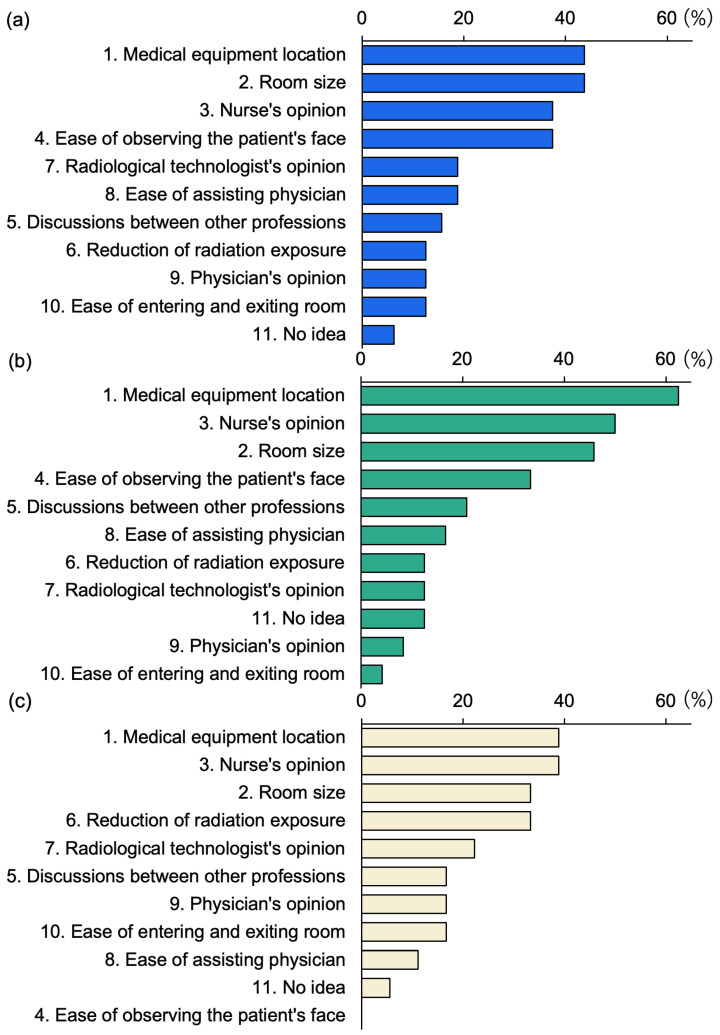
Positions (directions) of cardiovascular IR nurses and factors influencing their selection. (**a**) R-Head (*n* = 32, shown in blue), (**b**) R-Body (*n* = 24, shown in green), (**c**) R-Foot (*n* = 18, shown in ivory). The numbers attached to each factor correspond to those in Table 4.

**Table 1 nursrep-15-00011-t001:** Question items, response methods, and options.

Question Item		Response Method	Options
Basic information on respondent and their affiliated institution			
Overview of affiliated institution	Hospital type by establishment body		Private hospitals, public/public and social insurance-related hospitals, national hospitals, university hospitals
	District location	Multiple choice	47 prefectures
	Emergency medical system	Multiple choice	Primary emergency designation, secondary emergency designation, tertiary emergency designation, no emergency designation
	Number of hospital beds	Descriptive	
	Annual number of cardiovascular IR procedures conducted	Multiple choice	<100 cases, 100–299 cases, 300–499 cases, 500–699 cases, 700–899 cases, ≥900 cases
Cardiovascular IR work status *			
Angiography room environment	Size of angiography room	Multiple choice	<40 m^2^, 40–49 m^2^, 50–59 m^2^, 60–69 m^2^, ≥70 m^2^, unknown
	CSS installation status	Multiple choice	Installed, installed in part of the room, not installed
	RS installation status	Multiple choice	Installed, installed in part of the room, not installed
	Medical equipment located outside of angiography room	Multiple choice (multiple choices allowed)	Emergency cart, medicine cabinet, IR catheter storage shelf, hygiene material (infusion route, etc.) storage shelf, activated clotting time (ACT) measurement device, nursing recording media (computer for electronic medical records, etc.), all in room and not applicable
Staff personnel composition ^†^	Proportion of procedures (diagnosis, treatment) conducted by only one physician	Multiple choice	None at all, 10–20%, 30–40%, approximately 50%, 60–70%, 80–90%, almost all
	Proportion of procedures (diagnosis, treatment) conducted by three or more physicians	Multiple choice	
	Proportion of procedures (diagnosis, treatment) conducted by only one IR nurse	Multiple choice	
	Proportion of procedures (diagnosis, treatment) attended by resident	Multiple choice	
Position of IR nurse	Distance (cm) ^‡^	Descriptive	
	Direction ^§^	Multiple choice	A-T with 360° of room centered on radiation field divided into 20 sections
Determining factor of position of IR nurse	What criteria were used to determine the position of the IR nurse?	Multiple choice (multiple choices allowed)	Physician’s opinion, nurse’s opinion, radiological technologist’s opinion, discussion between other professions, room size, item arrangement, ease of entering and exiting room, ease of assisting operator, ease of observing patient, reduction in radiation exposure for nurses, unknown

CSS: ceiling-suspended lead shield. RS: rolling lead shield. * Participants were asked to respond about the angiography room that is most frequently used in cardiovascular IR. ^†^ Numbers do not include changes during the session. ^‡^ Straight-line distance from the center of rotation of the C-arm (X-ray irradiation field) to the location where the IR nurse recorded information. ^§^ The direction of the position of the IR nurse as seen from the X-ray irradiation field.

**Table 2 nursrep-15-00011-t002:** Comparison of IR staff composition by number of hospital beds.

Number of IR Staff				≤343 Beds (*n* = 26)	344–520 Beds (*n* = 27)	≥521 Beds (*n* = 25)	*p* (Kruskal–Wallis)	Multiple Comparisons (Mann–Whitney *U*)
IR procedure for diagnosis								
	Physician							
		one	mean ± SD	3.7 ± 2.5	3.2 ± 2.1	1.9 ± 1.4	0.02	≥521 beds vs. ≤343 beds *
			median [IQR]	2.5 [1–6.75]	2 [2–5]	2 [1–2]		≥521 beds vs. 344–520 beds *
		three or more	mean ± SD	2.1 ± 1.6	1.9 ± 1.2	2.8 ± 1.5	0.018	≤343 beds vs. ≥521 beds **
			median [IQR]	2 [1–2]	2 [1–2]	2 [2–4]		344–520 beds vs. ≥521 beds **
	Resident ^†^							
		one or more	mean ± SD	2.9 ± 1.5	3.5 ± 1.6	5.1 ± 1.8	<0.001	≤343 beds vs. ≥521 beds **
			median [IQR]	2 [2–4]	3 [2.5–5]	6 [3–7]		344–520 beds vs. ≥521 beds **
	IR nurse							
		one	mean ± SD	3.9 ± 2.3	5.0 ± 2.3	5.0 ± 2.0	0.099	
			median [IQR]	3.5 [2–6]	6 [3–7]	6 [4–7]		
IR procedure for treatment								
	Physician							
		one	mean ± SD	2.3 ± 2.1	2.2 ± 1.8	1.2 ± 0.4	0.092	
			median [IQR]	1 [1–2]	1 [1–2.5]	1 [1–1]		
		three or more	mean ± SD	2.8 ± 1.7	2.7 ± 1.3	4.3 ± 1.4	<0.001	≤343 beds vs. ≥521 beds **
			median [IQR]	2 [2–3.75]	2 [2–3.5]	4 [3–5]		344–520 beds vs. ≥521 beds **
	Resident ^†^							
		one or more	mean ± SD	2.9 ± 1.5	3.6 ± 1.7	4.8 ± 1.9	0.003	≤343 beds vs. ≥521 beds **
			median [IQR]	2 [2–4]	3 [3–4.75]	5 [3–7]		344–520 beds vs. ≥521 beds *
	IR nurse							
		one	mean ± SD	3.5 ± 2.4	4.3 ± 2.2	4.4 ± 2.4	0.375	
			median [IQR]	3 [1–6]	5 [2–6]	5 [2–7]		

Numbers are divided into seven levels: 1—not at all; 2—10–20%; 3—30–40%; 4—approx. 50%; 5—60–70%; 6—80–90%; 7—almost all. SD: standard deviation. IQR: interquartile range [1st–3rd quartile]. * *p* < 0.05, ** *p* < 0.01. ^†^ Physicians who have had their medical license for less than five years.

**Table 3 nursrep-15-00011-t003:** Comparison of other characteristics by number of hospital beds.

		≤343 Beds (*n* = 26)		344–520 Beds (*n* = 27)		≥521 Beds (*n* = 25)		*p* (χ^2^-Test)
		n (exp)	Adjusted *R*	n (exp)	Adjusted *R*	n (exp)	Adjusted *R*	
Hospital type	Private hospital	15 (11.7)	1.6	14 (12.1)	0.9	6 (11.2)	−2.5	0.017
	National/public hospital	9 (10.0)	−0.5	11 (10.4)	0.3	10 (9.6)	0.2	
	University hospital	2 (4.3)	−1.5	2 (4.5)	−1.6	9 (4.2)	3.1	
Emergency medical system ^#^	Primary emergency	4 (1.7)	2.3	0 (1.7)	−1.7	1 (1.6)	−0.6	<0.001
	Secondary emergency	17 (15.7)	0.7	24 (16.3)	3.8	6 (15.1)	−4.5	
	Tertiary emergency	4 (8.3)	−2.2	3 (8.7)	−2.9	18 (8.0)	5.2	
	No emergency designation	1 (0.3)	1.4	0 (0.3)	−0.7	0 (0.3)	−0.7	
Annual number of cardiovascular IR procedures conducted	<300 cases	13 (7.0)	3.2	8 (7.3)	0.4	0 (6.7)	−3.7	<0.001
	300–499 cases	5 (4.7)	0.2	6 (4.8)	0.7	3 (4.5)	−0.9	
	500–699 cases	4 (3.7)	0.2	5 (3.8)	0.8	2 (3.5)	−1.1	
	≥700 cases	4 (10.7)	−3.3	8 (11.1)	−1.5	20 (10.3)	4.8	
Size of angiography room ^†^	<50 m^2^	3 (5.3)	−1.4	6 (5.6)	0.3	8 (6.2)	1.1	0.390
	50–59 m^2^	7 (4.3)	1.8	4 (4.6)	−0.4	3 (5.1)	−1.3	
	60–69 m^2^	5 (5.3)	−0.2	7 (5.6)	0.9	5 (6.2)	−0.7	
	≥70 m^2^	2 (2.2)	−0.1	1 (2.3)	−1.1	4 (2.5)	1.2	
Medical equipment location ^‡^	Some placed outside room	13 (16.0)	−1.5	14 (16.6)	−1.3	21 (15.4)	2.8	0.020
	All placed in room	13 (10.0)	1.5	13 (10.4)	1.3	4 (9.6)	−2.8	
CSS	All placed in room	22 (24.3)	−2.3	27 (25.3)	1.7	24 (23.4)	0.6	0.115
	Some placed in room	1 (0.7)	0.5	0 (0.7)	−1.0	1 (0.6)	0.6	
	No placement	3 (1.0)	2.5	0 (1.0)	−1.3	0 (1.0)	−1.2	
RS	All placed in room	20 (21.0)	−0.6	22 (21.8)	0.1	21 (20.2)	0.5	0.177
	Some placed in room	1 (2.7)	−1.3	4 (2.8)	1.0	3 (2.6)	0.3	
	No placement	5 (2.3)	2.2	1 (2.4)	−1.2	1 (2.2)	−1.1	

n: number of cases. exp: expected number of cases. Adjusted *R*: standardized residual. If the absolute value is greater than 1.96, the item is significantly different at the 5% level. An absolute value of the residual larger than 2.58 indicates a significant difference at the 1% level. A positive (negative) value for the residual means that the item is significantly more (less) than the other items. ^#^ Primary (initial) emergency: for patients who can return home. Secondary emergency: for patients who require hospitalization or surgery. Tertiary emergency (emergency medical center): accepting patients with serious, life-threatening conditions 24 h a day. ^†^ Responses from 55 facilities were analyzed, with the 23 facilities that responded “unknown” excluded. ^‡^ Six items: emergency cart, medicine cabinet, IR catheter storage shelf, hygiene material (infusion route, etc.) storage shelf, activated clotting time (ACT) measurement device, nursing recording media (computers for electronic medical records, etc.). CSS: ceiling-suspended lead shield; RS: rolling lead shield.

**Table 4 nursrep-15-00011-t004:** Factors determining position of IR nurses.

Option	Number of Selecting Hospitals (%)
1. Medical equipment location	36 (46.2)
2. Room size	32 (41.0)
3. Nurse’s opinion	31 (39.7)
4. Ease of observing the patient’s face	21 (26.9)
5. Discussions between other professions	15 (19.2)
6. Reduction in radiation exposure	14 (17.9)
7. Radiological technologist’s opinion	13 (16.7)
8. Ease of assisting physician	12 (15.4)
9. Physician’s opinion	9 (11.5)
10. Ease of entering and exiting room	8 (10.3)
11. No idea	8 (10.3)

## Data Availability

The data presented in this study are available at the request of the corresponding author due to the study’s aims and ethics.

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
