# Peer review of "Influence of Hospital Bed Count on the Positioning of Cardiovascular Interventional Radiology (IR) Nurses: Online Questionnaire Survey of Japanese IR-Specialized Radiological Technologists"

_nursrep, 2025, doi:10.3390/nursrep15010011_

Round 1
Reviewer 1 Report
Comments and Suggestions for Authors
Topic: Influence of hospital bed count on the positioning of cardio vascular interventional radiology (IR) nurses: Online question naire survey of Japanese IR-specialized radiological technologists
Abstract: Accept
Introduction:
· In the Introduction section, should research related to how radiation levels in medical settings may affect living organisms be added to enhance the study's interest and significance
· Should definitions of 'ceiling suspended lead shields (CSS)' and 'rolling shields (RS)' be added to the introduction to make the content more accessible to the general audience? Additionally, would it be possible to include an example image or labels in Figure 2 to enhance clarity
Materials and methods:
· In line 75, where it states 'questionnaire developed by the researcher,' should the process of developing the questionnaire be elaborated to include credible methods, such as involving experts or applying the IOC (Index of Item-Objective Congruence) principle? This is because IOC is a critical step in developing high-quality questionnaires, ensuring that the questions align with the research objectives.
· For the online survey process, could you provide more details about whether it was conducted through a chat system or email? Complete information should be included.
· In line 91, where it states 'The survey period extended from February 1, 2023, to November 30, 2023,' should a justification be provided to explain why this time frame is reliable?
Results
· How reliable is the sample size of data from 78 facilities in this study? If fewer facilities had been included, how would it have impacted the validity and outcomes of the research?
Discussion
· Should related or contrasting research studies be included, along with a clear description of the unique contributions of this study, to highlight its potential for global application?
Conclusion: Accept
Comments on the Quality of English LanguageThe manuscript would benefit from a thorough grammar check to ensure clarity and precision.
Author Response
Introduction:
- In the Introduction section, should research related to how radiation levels in medical settings may affect living organisms be added to enhance the study's interest and significance.
Thank you for your valuable comments. We added the phrases in the Introduction section (Line 54): During typical fluoroscopic working conditions, the cumulative exposure of interventionalists, other physicians, and/or medical staff working close to the patient, even if the exposure dose per procedure is low, can be high due to repeated procedures, and there is a concern that the risk of radiation injuries will increase. There have been many reports of the high risk of developing lens opacity pre-cataracts among medical staff working in IR [a-j]. And now, the threshold dose for the eye lens is thought to be 0.5 Gy [k], and the lens is considered one of the most radiation-sensitive tissues in the body [l,m].
[a] Vañó, E., Gonzalez, L., Beneytez, F., et al., 1998. Lens injuries induced by occupational exposure in non − optimized interventional radiology laboratories. Br. J. Radiol. 71, 728−733.
[b] Junk, A.K., Haskal, Z., Worgul, B.V., 2004. Cataract in interventional radiology―an occupational hazard? Invest. Ophthalmol.Vis. Sci. 4(5 Suppl. S 2005), 388.
[c] Chodick, G., Bekiroglu, N., Hauptmann, M., et al., 2008. Risk of cataract after exposure to low doses of ionizing radiation : a 20−year prospective cohort study among US radiologic technologists. Am. J. Epidemiol. 168, 620−631.
[d] Kleiman, N.J., Cabrera, M., Duran, G., Ramirez, R., Duran, A., Vañó, E., 2009. Occupational risk of radiation cataract in interventional cardiology. Invest. Ophthalmol. Vis. Sci. 49, Presentation abstract 511/D 656.
[e] Vano, E.; Kleiman, N.J.; Duran, A.; Rehani, M.M.; Echeverri, D.; Cabrera, M. Radiation cataract risk in interventional cardiology personnel. Radiat. Res. 2010, 174, 490–495.
[f] Ciraj-Bjelac, O.; Rehani, M.M.; Sim, K.H.; Liew, H.B.; Vano, E.; Kleiman, N.J. Risk for radiation-induced cataract for staff in interventional cardiology: Is there reason for concern? Catheter. Cardiovasc. Interv. 2010, 76, 826–834.
[g] Rehani, M.M.; Vano, E.; Ciraj-Bjelac, O.; Kleiman, N.J. Radiation and cataract. Radiat. Prot. Dosim. 2011, 147, 300–304.
[h] Jacob, S., Boveda, S., Bar, O., et al., 2012. Interventional cardiologists and risk of radiation induced cataract: results of a French multicenter observational study. Int. J. Cardiol. 167, 1843–1847.
[i] Ciraj-Bjelac, O.; Rehani, M.; Minamoto, A.; Sim, K.H.; Liew, H.B.; Vano, E. Radiation-induced eye lens changes and risk for cataract in interventional cardiology. Cardiology 2012, 123, 168–171.
[j] Vano, E.; Kleiman, N.J.; Duran, A.; Romano-Miller, M.; Rehani, M.M. Radiation-associated lens opacities in catheterization personnel: Results of a survey and direct assessments. J. Vasc. Interv. Radiol. 2013, 24, 197–204.
[k] International Commission on Radiological Protection. ICRP statement on tissue reactions and early and late effects of radiation in normal tissues and organs—Threshold doses for tissue reactions in a radiation protection context. Ann. ICRP 2012, 41, 1–322.
[l] Brown, N.P., 1997. The lens is more sensitive to radiation than we had believed. Br. J. Ophthalmol. 81, 257.
[m] Ainsbury, E.A., Bouffler, S.D., Dorr, W., et al., 2009. Radiation cataractogenesis : a review of recent studies. Radiat. Res. 172, 1−9.
- Should definitions of 'ceiling suspended lead shields (CSS)' and 'rolling shields (RS)' be added to the introduction to make the content more accessible to the general audience? Additionally, would it be possible to include an example image or labels in Figure 2 to enhance clarity.
Thank you for your comments. We added some words to define the CSS and RS and inserted a new Figure 1 in the Introduction section.
(Line 58) Previous studies have also identified factors affecting the radiation dose to IR nurses, including their position relative to the patient and X-ray tube [4,5,6]; their role [6,7,8,9]; the use of shielding devices, such as ceiling-suspended lead shields (CSS) and rolling shields (RS), to protect physicians and other staff, respectively [6,7,10,11,12]; and their calling out to the operator before approaching the patient [13].
Materials and methods:
- In line 75, where it states 'questionnaire developed by the researcher,' should the process of developing the questionnaire be elaborated to include credible methods, such as involving experts or applying the IOC (Index of Item-Objective Congruence) principle? This is because IOC is a critical step in developing high-quality questionnaires, ensuring that the questions align with the research objectives.
Thank you for your valuable feedback. Unfortunately, the IOC was not implemented in this study. However, the validity of the questions and their contents was examined by experts in the IR field. The breakdown of the five people who made the questions has been added.
(Line 105) The questionnaire was developed by a team of five, including four IR-specialized radiological technologists and one nurse dedicated to researching radiation protection for medical workers.
- For the online survey process, could you provide more details about whether it was conducted through a chat system or email? Complete information should be included.
Thank you for your comments. In line 83, we added, “The participants accessed the relevant website of their own volition.”
- In line 91, where it states 'The survey period extended from February 1, 2023, to November 30, 2023,' should a justification be provided to explain why this time frame is reliable?
Thank you very much for pointing out our mistake. It should have been, “The survey was conducted from February 1, 2023 to November 30, 2023.” We have corrected it (Line 91).
Results
- How reliable is the sample size of data from 78 facilities in this study? If fewer facilities had been included, how would it have impacted the validity and outcomes of the research?
Thank you for your comments. Since the target population size was 434, the sampling error of the data from the 78 facilities was about 10% (confidence level 95%). We will add this information to the Results. In addition, this survey aimed to verify the hypothesis derived from the survey results of three facilities (all of which had a radiological technologist specializing in interventional radiology (IR)) in the previous report [Kuriyama, 2024]. When the 78 facilities were divided into three groups based on the number of beds and the differences between the groups were analyzed, a clear statistically significant difference was shown. In that sense, the reliability of the required data has been met at a minimum. However, we do not believe this can be considered a trend for all medical facilities in Japan with angiography equipment, of which there are more than 1,500, so we will state this in the Limitations.
We rewrote the phrase (Line 142): “This represents approximately 18% of the 434 facilities in Japan where IR-specialized radiological technologists are employed; thus, the data's sampling error could be estimated to be approximately 10% (confidence level 95%).”
We rewrote the phrase (Line 433): Although we surveyed a significant number of hospitals (78 hospitals), this cannot be considered a trend for all medical facilities in Japan with angiography equipment, of which there are more than 1,500. There is a possibility that the responses primarily came from IR-specialized radiological technologists who have a strong interest in radiation protection, which could introduce an online bias.
Discussion
- Should related or contrasting research studies be included, along with a clear description of the unique contributions of this study, to highlight its potential for global application?
Thank you for your valuable comments. We added the phrases in the Discussion section (Line 428): Personal dose monitoring is essential for assessing the exposure risk of IR nurses. However, in the world, there are still many countries and regions where personal dosimeters are unavailable or not worn daily [n,o]. There are also reports that the use rate of personal dosimeters is higher among nurses than among physicians [j,p], but in the case of passive dosimeters, the cumulative dose for a certain period is notified to the user, so it isn't easy to use the dose results to improve the situation. Even in this situation, the number of hospital beds is a dose indicator that can be obtained easily without spending money, so it is expected to be used for risk assessment of IR nurses.
[n]International Commission on Radiological Protection. Occupational Radiological Protection in Interventional Procedures. Ann. ICRP 2018, 47, 1–118.
[o] van der Merwe, B. Establishing ionising radiation safety culture during interventional cardiovascular procedures. Cardiovasc. J. Afr. 2021, 32, 271–27.
[j] Vano, E.; Kleiman, N.J.; Duran, A.; Romano-Miller, M.; Rehani, M.M. Radiation-associated lens opacities in catheterization personnel: Results of a survey and direct assessments. J. Vasc. Interv. Radiol. 2013, 24, 197–204.
[q] Ministry of Health, Labour and Welfare. A study on the actual status of radiation exposure in radiation work involving uneven exposure and the assessment of issues for reducing radiation exposure; Research Summary Report: FY2020 industrial accident disease clinical research project. https://www.mhlw.go.jp/content/000812185.pdf (accessed on 22 Dec 2024).
Reviewer 2 Report
Comments and Suggestions for Authors
You stated that “hospitals with fewer beds tended to have a shorter distance between the X-ray irradiation field and the IR nurse's position”, my questions are:
Are you correlating the size of the hospitals with the number of beds? If so, it is possible that a modern hospital is usually spacious, but still is considered a smaller hospital based on the number of beds.
Also, how do you qualify the distance between the X-ray irradiation field and the IR nurse's position?
I would like to see a threshold identified as far as the distance of the radiation field based on the intensity of the radiation.
Also, please identify how you correlate the size of the hospital with the number of beds and the distance from the radiation field. Could you also report correlation coefficients?
Considering shorter distances in smaller hospitals, how many X-ray procedures should the providers perform in a small hospital per month instead of a large one?
Also, specify what you consider small and large hospitals early in your abstract.
Author Response
You stated that “hospitals with fewer beds tended to have a shorter distance between the X-ray irradiation field and the IR nurse's position”, my questions are:
- Are you correlating the size of the hospitals with the number of beds? If so, it is possible that a modern hospital is usually spacious, but still is considered a smaller hospital based on the number of beds.
As you pointed out, this paper analyzes the data assuming that hospital size correlates with the number of hospital beds. In Japan, hospital size is evaluated according to the number of beds in national statistical data, and staffing standards are determined based on this classification. The concept of hospital size does not consider the exclusive area of patient rooms or operating rooms. As this way of thinking and classification may be unique to Japan, we have reviewed the entire text to avoid subjective qualitative expressions such as “smaller hospital” and “larger hospital” as much as possible and to express it in quantitative terms using the number of beds, so as not to cause misunderstanding. Or, when you say that modern hospitals are spacious, are you referring to the size of the hospital building or the size of the grounds? If so, please understand that we have not analyzed the size of the hospital grounds or the total floor space in this paper. Furthermore, we do not have information on whether hospitals with large buildings or grounds a large or small number of beds have, so we cannot respond appropriately.
- Also, how do you qualify the distance between the X-ray irradiation field and the IR nurse's position?
The distance was defined as the straight-line distance between the X-ray irradiation field and the IR nurse's position. We clearly stated this in the footnotes of Table 1.
- I would like to see a threshold identified as far as the distance of the radiation field based on the intensity of the radiation.
The data from this study does not provide sufficient numerical information to establish a threshold distance for the position of the IR nurse. However, our previous report demonstrated the relationship between the equivalent dose rate of the crystalline lens of the IR nurse and the distance between the X-ray irradiation field and the nurse's position. It indicated that the slope of the regression curve becomes less steep when the distance exceeds 2 m, suggesting that at least 2 m from the X-ray irradiation field is the appropriate distance for the IR nurse, leading to an effective reduction in the equivalent dose to the crystalline lens (Kuriyama, 2024, Figure 4).
- Also, please identify how you correlate the size of the hospital with the number of beds and the distance from the radiation field. Could you also report correlation coefficients?
Thank you for your comments. The data from the correlation analysis mentioned was added in 3.1 Comparison of distance between position of the IR nurse and X-ray irradiation field by number of hospital beds section. The correlation coefficient was r = 0.364.
(Figure Legends) Figure 1. Association between the number of hospital beds and the distance between X-ray irradiation field and the position of the IR nurse. The solid line shows the regression line. r2, coefficient of determination.
- Considering shorter distances in smaller hospitals, how many X-ray procedures should the providers perform in a small hospital per month instead of a large one?
Thank you for your question. This study did not analyze individual radiation doses, so we cannot determine how often X-ray examinations are acceptable. However, as long as individual radiation doses do not exceed the limit, and the air dose at the boundary of the radiation-controlled area remains within this limit, X-ray examinations can be performed as often as necessary.
- Also, specify what you consider small and large hospitals early in your abstract.
Thank you for your comment. As with 1), we have revised the entire text to avoid subjective qualitative expressions such as “smaller hospital” and “larger hospital” that could lead to misunderstanding and to express the number of beds quantitatively.
Round 2
Reviewer 1 Report
Comments and Suggestions for Authors
I have reviewed the revised version of the manuscript titled "Influence of hospital bed count on the positioning of cardiovascular interventional radiology (IR) nurses: Online questionnaire survey of Japanese IR-specialized radiological technologists." Based on the revisions made, I find the paper to be acceptable for publication.